# Correlation between Mild Cognitive Impairment and Sarcopenia: The Prospective Role of Lipids and Basal Metabolic Rate in the Link

**DOI:** 10.3390/nu14245321

**Published:** 2022-12-15

**Authors:** Xuan Wang, Rong Xiao, Hongrui Li, Tiantian Li, Lizheng Guan, Huini Ding, Xiaoying Li, Huaguang Zheng, Kang Yu, Ai Zhao, Wannian Liang, Yuandi Xi

**Affiliations:** 1Beijing Key Laboratory of Environmental Toxicology, School of Public Health, Capital Medical University, Beijing 100069, China; 2Beijing Jishuitan Hospital, Beijing 100035, China; 3Beijing Tiantan Hospital, Capital Medical University, Beijing 100050, China; 4Peking Union Medical College Hospital, Beijing 100730, China; 5Wanke School of Public Health, Tsinghua University, Beijing 100190, China

**Keywords:** mild cognitive impairment, sarcopenia, dietary fats, lipid profile, lipidomics

## Abstract

There is evidence of correlation between mild cognitive impairment (MCI) and sarcopenia (SA). However, the influencing factors and the mechanism, such as age-related lipid redistribution, remain unknown. This study aimed to clarify the role of dietary fats and erythrocyte lipids profile combined with basal metabolic rate (BMR) in the link between MCI and SA. A total of 1050 participants aged 65 to 85 were divided into control, MCI, SA and MCI and SA groups. Bioelectrical impedance analysis was used to evaluate appendicular lean mass and BMR. Cognition and dietary nutrition were detected by neuropsychological tests and food frequency questionnaires. UHPLC-QExactive-MS/MS and UHPLC-Qtrap-MS/MS were used to conduct the lipidomics analysis. Lower dietary intake of different phospholipids, unsaturated fatty acids and kinds of choline were significantly associated with MCI and SA. Least absolute shrinkage and selection operator, multivariate logistic regression, receiver operating characteristic curve and validation tests provided evidence that specific phospholipids, unsaturated fatty acids and BMR might be the critical factors in the processing of MCI and SA, as well as in their link. The lipidomic analysis observed a clear discrimination of the lipid profiles in the individuals who are in MCI, SA, or MCI and SA, compared with the control. Lower expressions in certain phospholipid species, such as sphingomyelin and phosphatidylethanolamines, decreased phosphatidylcholine with more unsaturated double bonds, lower level of lipids with C20:5 and C20:4, higher level of lipids with C18:2 and lipids with a remodeled length of acyl chain, might be closely related to the link between MCI and SA. Inadequate dietary intake and lower concentrations of the erythrocyte lipid profile of phospholipids and unsaturated fatty acids with a lower level of BMR might be the key points that lead to progress in MCI and SA, as well as in their link. They could be used as the prospective biomarkers for the higher risk of cognitive decline and/or SA in elderly population.

## 1. Introduction

Cognitive impairment and sarcopenia (SA) are two of the most prevalent causes of disability in the aging population. The loss of independence induced by these mental and physical dysfunctions could seriously affect the quality of life in the elderly and bring a severe problem of medical costs.

Mild cognitive impairment (MCI) is recognized as a transitional stage between healthy aging and dementia. Evidence shows that 15 to 38% of MCI patients older than 65 will develop dementia within the following 2 to 5 years. The failure of drug trials in Alzheimer’s disease (AD) treatment has turned researchers’ focus to identifying which groups of individuals are more likely to develop MCI/AD.

SA is the degenerative loss of skeletal muscle mass and strength, which increases the risk of falls, physical impairment, poor quality of life and mortality in elderly people. Increased studies illustrate the crosstalk between SA and MCI. Evidence shows that participants with MCI/AD have a high prevalence rate of SA [1,2]. Another piece of research with 297 participants over 65 years old finds that low short physical performance battery score is associated with a 2.22-fold higher risk of cognitive impairment [3].

Despite the association between MCI and SA being quantified, the influencing factors and the mechanism of their relationship remain unknown. Only a few researchers have speculated that the link might be associated with a particular sort of lifestyle, reduced appetite, or malnutrition [2]. Our previous cognitive cohort study finds MCI individuals are more likely to be troubled by decreased muscle mass or strength [4,5]. Moreover, such people tend to have a lipid metabolism disorder, such as cholesterol, triglycerides and phospholipids. In this study, we aimed to clarify the correlation between MCI and SA and tried to further investigate the role of dietary fats and erythrocyte lipid profile combined with basal metabolic rate (BMR) in the link through lipidomic detection.

## 2. Materials and Methods

### 2.1. Participants

In Beijing from 2020 to 2021, a group of participants between 65 to 85 years old were sampled (registered at Chinese Clinical Trial Registry as ChiCTR2100054969). The workflow and standards were adopted as in a previous study [4,5]. Finally, 1050 participants entered the study who had signed informed consent. This work was conducted in accordance with the principles of the Declaration of Helsinki and ethically approved by the Ethics Committee of Capital Medical University (Z2019SY052). 

### 2.2. Sarcopenia Assessment

Sarcopenia is assessed based on the (Asian Working Group for Sarcopenia) AWGS 2019 criteria, defined as participants with low SMI and low grip strength and/or low physical function [6,7,8,9]. The details can be found in Appendix A.

### 2.3. Cognitive Assessment

Cognitive function was assessed by the Montreal cognitive assessment (MoCA) and mini-mental state examination (MMSE) score [10]. The two-stage procedure to diagnose MCI patients refers to our previous study [5,11].

### 2.4. Dietary Assessment

The dietary information was collected by food frequency questionnaire (FFQ) of 2002 China National Nutrition and Health Survey (CNHS 2002) [12]. Energy and nutrients intake were calculated by using the China Food Composition Database (Version 6) [11,13].

### 2.5. Physical Activity Evaluation

Participants performed the self-report of physical activity scale for the elderly (PASE) [14], which is suitable for people older than 65 during the initial and follow-up phases of the study.

### 2.6. Blood Sample Collection

Blood samples were collected in the morning from fasting participants who fasted from eight the night before. Serum cholesterol levels, which include total cholesterol (TC), triglyceride (TAG), high-density lipoprotein cholesterol (HDL-C) and low-density lipoprotein cholesterol (LDL-C), were measured by enzymatic method in automatic biochemistry analyzer (Olympus AU480, Japan).

### 2.7. Nontargeted Lipidomics

The method of erythrocyte lipid profile detection is similar to the previous study [15]. Lipids extraction, a modified method, was employed. The details of lipids extraction and the information of reagents, processes and acquisition software could be found in Appendix A.

### 2.8. MRM Targeted Measurement

In this study, the experimental method is the same as the previous study [15]. Preliminary treatment of erythrocyte was consistent with the protocol of untargeted lipidomics. Skyline 20.1 software was employed for the quantification of the target compounds. The absolute content of individuals’ lipids, corresponding to the internal standard (IS), was calculated on the basis of peaks area and the actual concentration of the identical IS lipid class, and then absolute content was obtained from diverse IS averages of the identical lipid class.

### 2.9. Statistical Analysis

Continuous variables were expressed as medians (interquartile ranges, IQR) when non-normally distributed or the mean ± standard deviation (SD) when normally distributed. Analysis of variance (ANOVA) or the Kruskal–Wallis rank test was applied for continuous variables, while chi-squared tests were used for categorical variables. Multiple linear regression analysis was used to examine the relationship of key factors. Least absolute shrinkage and selection operator (LASSO) was used to select the predictor variables for multifactor logistic regression models [5,13,16]. Receiver operating characteristic curve (ROC) was carried out to compare the efficiency of BMR in each prediction model [17]. After that, participants were randomly divided into training and validation groups at a ratio of 3:1 for the further validating of each model [13,16,18]. ROC, Hosmer–Lemeshow tests and calibration curves [18] were used to test the consistency of predicted probabilities and observed frequencies. In addition, the nomogram was constructed to show the prediction model more intuitively [18]. Statistical significance was set at a two-sided *p* < 0.05. All statistical analyses were performed using R studio software programs and IBM SPSS Statistics 26. The software GraphPad Prism 8 was employed for the box and bar plot.

## 3. Results

### 3.1. Demographic and Clinical Characteristics of Participants

As shown in Table 1, there were significant differences in age, body mass index (BMI) and BMR, in comparison with SA vs. control and MCI and SA vs. control. Therefore, the patients of SA and MCI and SA were older, while BMI and BMR were much lower than the individuals of control and MCI people. In addition, poorly educated people were at higher risk of SA, while people with higher education than junior high school were more likely to have MCI. As expected, MoCA score was definitely lower in all individuals of MCI, SA, and MCI and SA groups than in the control.

### 3.2. Comparison of Dietary Consumption in Different Groups

As shown in Table 1, the obvious discrepancies of dietary lipids were shown in different groups. In MCI and SA patients, the dietary intake of phosphatidylcholine (PC) and sphingomyelin (SM), as well as polyunsaturated fatty acid (PUFA) and monounsaturated fatty acid (MUFA) were significantly lower than in controls. It was noteworthy that PUFA and saturated fatty acid (SFA) were the lowest in MCI and SA patients out of the groups. In addition, the consumption of betaine, glycerol-phosphatidylcholine (GPC) and phosphorylcholine (PCho) were significantly lower in MCI and SA patients, while the consumption of betaine and PCho were also lower in MCI individuals than in controls.

### 3.3. Performance of BMR and Dietary Consumption on MoCA Score and Skeletal Muscle Index

First, in Table 2, the higher MoCA score was closely associated with higher skeletal muscle index (SMI) in both model 1 (adjusted for age, sex, education, BMI, TAG and Arthritis) and model 2 (adjusted for the variables in model 1 and PASE score). 

Importantly, BMR showed a surprising correlation with the MoCA score and SMI in all tests of model 1 and model 2. As BMR is influenced by daily diet, physical activity and age, it might be a vital factor affecting the processes and the link between MCI and SA. More interestingly, SMI was brought into the regression equation of the MoCA score and could make the power of BMR disappear. It implied a very strong correlation between the SMI and MoCA score [19].

Furthermore, a higher dietary consumption of PC, SM, SFA, MUFA, PUFA, betaine, GPC, PCho and cholesterol was significantly related to a higher MoCA score in both model 1 and model 2. However, the intake of fat and total fatty acids could be found negatively correlated with SMI, and the intake of fat negatively correlated with the MoCA score. The positive correlation of protein and negative correlation of CHO in SMI could disappear after a more adjusted PASE score (model 2). 

### 3.4. The Role of Dietary Fats and BMR in the Relationship between MCI and SA

*Variables selection*—A total of 63 variables (Appendix A) were incorporated into every LASSO model [13]. Variables were filtered out based on the lambda min. value, corresponding to the smallest loss (Figure 1) [16]. The selected factors in each group were shown in Appendix A. These variables had non-zero coefficients and could be used to screen the risk factors of MCI, SA, or MCI and SA [17].

*Multivariate logistic regression analysis*—As shown in Table 3, in the model of MCI–control, Q3 of BMR, as well as Q3 and Q4 of PCho were associated with a reduced risk of MCI after adjusting demographic and clinical parameters. In SA–control, Q2, Q3 and Q4 of BMR and Q2 of C20:1 were associated with a reduced risk of SA. In MCI and SA–control, Q2, Q3 and Q4 of BMR, as well as Q4 of protein and Q4 of C22:1 were associated with a reduced risk of MCI and SA.

*The role of BMR in prediction model*—The variables screened by logistic regression were used to conduct ROC prediction models (Figure 1). Results showed that the models yielded a better improvement in the prediction of MCI, SA, and MCI and SA when including BMR.

*Training and validation sets*—Participants were separated at a ratio of 3:1 to complete the Hosmer–Lemeshow test, calibration chart and new ROC analysis [13]. The ROC results indicated a very good performance of the models (Figure 1). It could also be proved by calibration curves that the prediction model had a good fit in the training and validation set, as well as the results of Hosmer–Lemeshow test demonstrating that the predicted probability was highly consistent with the actual probability (Figure 1).

### 3.5. Nomogram for the Prediction Models of Multivariate Logistic Regression

Finally, nomograms were used for quantitative prediction of the risk probability of developing MCI, SA, and MCI and SA (Figure 1) [17]. In the nomogram of MCI–control, a person in the primary school education while in Q1 of BMR and in Q3 of PCho had a 43% risk of MCI; in SA–control, a male in Q1 of BMR and in Q2 of C20:1 had an 80% risk of SA; in MCI and SA–control, a female between the ages of 65 to 69, with Q1 of BMR, Q3 of protein intake, or Q3 of C22:1 intake, with arthritis but without hypertension, had a 65% risk of MCI and SA.

### 3.6. Nontargeted Lipidomics Analysis

Erythrocyte lipid profile was a reflection of long-term dietary lipid intake. The differences in the dietary phospholipid and fatty acids, as shown above, made us believe that there might be changes in the erythrocyte lipid profile in MCI and/or SA. In that, nontargeted and targeted lipidomics analyses were conducted to explore the potential biomarkers of the erythrocyte lipids of participants.

*Global lipidomics analysis*—To investigate whether the erythrocyte lipid composition differs in four groups, 20 samples of control, 30 samples of MCI, 30 samples of SA, 15 samples of MCI and SA were detected by UHPLC-QE-MS. A total of 297 structures of lipids in negative ionization mode, as well as 404 structures in positive ionization mode, including glycerolphospholipids, sphingolipids, acylglycerolipids, saccharolipids, fatty acids esters and free fatty acids, were disclosed and identified. A typical total ion chromatogram (TIC) of the pooled positive and negative quality control (QC) sample from MCI–control, SA–control, and MCI and SA–control were provided in the Appendix A.

*Erythrocyte lipid profiles altered in each comparison*—Unsupervised principle components analysis (PCA) was conducted in this study to reveal the cases (MCI, SA, and MCI and SA)-related lipidomics discrepancies. The results showed obvious trends of separation and clustering separately in Appendix A. In addition, supervised orthogonal partial least squares discriminant analysis (OPLS-DA) was used to confirm the cases-related lipidomics discrepancies. As was shown in Appendix A, the OPLS-DA score plot revealed a clear separation in all three comparisons. Furthermore, permutation tests were conducted to prevent the overfitting of the models. Appendix A showed that the model had good predictability and did not overfit.

*The most prominent changes in lipid in each comparison*—Bubble plots, based on each comparison, were used to detect the distribution of subclasses in every lipid category (Appendix A). A total of 35 upregulated and 11 downregulated lipids in MCI–control, 61 upregulated and 6 downregulated lipids in SA vs. control, and 68 upregulated and 11 downregulated lipids in MCI and SA–control were found significantly changed in positive ionization mode, and the results of negative ionization are shown in Figure 2.

Heat maps were shown to be the most significantly disordered lipids (Appendix A). In the positive ionization mode, PC, TAG and SM were the top three changing subclasses in all three comparisons. Ceramides (Cer) were altered more significantly in only the SA vs. control and MCI and SA vs. control. In addition to that, in the negative ionization mode, PC, phosphatidylethanolamines (PE), phosphatidylserine (PS) and Cer were the main altered lipids in all three comparisons. The SM was altered more obviously in both SA–control and MCI and SA–control. It implied that PC, PE, SM and TAG might be the key altered subclasses in the link between MCI and SA. PE and SM might be easier to trigger than SA.

*The lipids with the consistent altered trend in three comparisons*—We sought to detect the lipids that had the same altered trend in three comparisons. An integrated evaluation based on univariate analysis (*p* value) and multivariate analysis (variable importance in the projection, VIP) was performed. Finally, 60 differential lipid species were screened (VIP > 1, *p* < 0.05) (Appendix A). Consistently upregulated lipids could be found, such as TAG, PC, PS, PE, SM, Ce, and so on. Most species of TAG and certain PC with the acyl chain of less than two unsaturated double bonds were consistently upregulated in all three comparisons.

### 3.7. MRM Targeted Measurement

*Altered lipids in each comparison*—In targeted lipidomics, positive and negative ionization modes were combined to make the lipidomics coverage higher and the detection effect better. A total of 350 distinct lipids, including glycerolphospholipids, sphingolipids, acylglycerolipids, saccharolipids, cholesterol esters and free fatty acids, were detected by UHPLC-QTRAP^®^ 6500^+^-MS/MS. The TIC of the pooled QC sample could be seen in Appendix A. The results of PCA, OPLS-DA and permutation tests were shown in Appendix A. The parameters of all the above results could be found in Appendix A, which revealed a clear separation between the cases and control.

Bubble plots were used to exhibit the distribution of discrepancy lipids in Appendix A. Heat maps gave the details of every differential lipid in three comparisons in Appendix A. Finally, 13 out of 350 differential lipids were found in MCI–control, 25 were found in SA–control and 16 were found in MCI and SA–control. It is noteworthy that the absolute value expressions of PE, SM and FFA were obviously lower in three case groups, compared with the control, while HexCer was significantly higher in all case groups. The classes of TAG were only overexpressed in MCI individuals, compared with controls.

*The quantitative expression of lipids changed in the consistent trend among three comparisons*—In Appendix A, PE (P-16:0/18:0), PE (P-16:0/18:1), PE (P-18:0/18:1), PE (P-18:1/18:1) and SM (26:0) were downregulated in all three case groups, compared with control, while HexCer (18:1/24:0) was the upregulated one. It implied that the downregulation of PEs and SMs were most likely the considerable biomarkers at risk of suffering from MCI and SA, as well as in their link.

The variation in PC was further developed. PC (18:2/20:4) was found to be definitely downregulated in the SA and MCI and SA individuals, compared with the control. Interestingly, PC (18:2/20:4), PE (18:2/20:4) and PE (P-18:2/20:4), which contain C20:4 (arachidonic acid, AA) and C18:2 (linoleic acid, LA) chain, as well as six unsaturated double bonds in total, declined in case groups (Appendix A. It turned our attention to the number of unsaturated double bonds and the type of lipid acid chains for the next step.

*The crucial role of acyl chain composition*—In Figure 3, PC with six double bonds significantly decreased in all three case groups, compared with the control. In addition, PE with six more double bonds manifested a decreasing trend in case groups, although no differences. The above results implied that the downregulated concentrations of specific PEs and PCs with more double bonds acyl chain in the erythrocyte might be correlated with a higher risk of suffering from MCI and SA, as well as their link.

Lipids with different types of acyl chain were detected in Appendix A. The FFA of MUFA and PUFA had no difference between the cases and control, while SFA significantly decreased in all case groups. Furthermore, the lipids with chains of C20:5 (eicosapentaenoic acid, EPA), C22:6 (docosahexenoic acid, DHA), AA and LA were calculated for the next step. Appendix A showed that the percentage of EPA-lipids, DHA-lipids and AA-lipids had a decreasing trend in case groups, compared with the control, although it had no differences. The percentage of LA-lipids was significantly upregulated in MCI vs. control and SA vs. control. 

Moreover, the discrepancies of the subclasses with EPA, DHA, AA and LA were in Appendix A. EPA-CE in the individuals of MCI and SA groups were significantly lower than that in the control. AA-PC and AA-CE in SA and MCI and SA were significantly lower. LA-CE in all case groups were higher than in the control. Free fatty acids were explored as well. In Appendix A, C16:0 (palmitic acid, PA), C18:0 (stearic acid, SA), C20:0 (arachidic acid, AA) and C18:3 (linolenic acid, LA) had differential declines in the case groups, compared to the control. However, the free fatty acids of DHA, AA and LA had no difference between cases and control.

The percentage of lipids with different lengths of acyl chain was detected next. In Appendix A, lipids with chain lengths of C16 and C18 manifested an increasing trend in case groups, while lipids with C22 and C26 showed decreasing trends. C24 could be downregulated approximately because of *p* = 0.052. The above results made lipids with EPA, AA and LA and lipids with longer-length chains than C24 be considered as im-portant factors in the link between MCI and SA.

## 4. Discussion

MCI and SA are recognized as the multifactorial syndromes that affect the aged population and contribute to their high mortality and poor quality of life [20]. Although the relationship of MCI and SA has been reported, there are still many unanswered questions in their link [21,22]. This study reports the correlation between MCI and SA from the perspective of dietary nutrition with different levels of BMR in elderly people, and nontargeted and targeted lipidomics are further detected to validate the results of dietary nutrition and specific lipids as biomarkers of MCI, SA and even their link.

In a previous study, there is a phenomenon in which MCI individuals are more likely to have decreased muscle mass and strength [1,2]. The present study indicated that the MoCA score was definitely lower in all individuals of MCI, SA, and MCI and SA groups, compared with control participants. Multiple linear regression analysis also gives evidence of a strong correlation between the MoCA score and SMI. This is similar to research that demonstrated how age-related muscle deterioration [23] and SA [24] are associated with an increasing risk of impaired cognitive function in the elderly. These made us believe that SA patients are more likely to develop MCI.

Demographic and clinical characteristics analysis indicated that higher age, lower education, lower BMI and lower BMR were the key factors in the progression and the relationship of MCI and SA. It was noteworthy that, compared with control group, the difference of BMR was not found in individuals of MCI but in SA and MCI and SA. These results implied BMR might have contributing factors to SA in MCI patients.

In order to evaluate the potential role of dietary lipids, the univariate analysis showed that the intake of phospholipids (PC, SM) and fat acids (MUFA, PUFA, SFA) were lower in MCI and SA patients. Choline is not only the metabolite of PC but also its synthetic material in human body. In present study, the choline of betaine and PCho were found in lower consumption in MCI and MCI and SA participants. These results suggested that PC-, SM-, MUFA-, PUFA-, SFA- and PC-related choline might contribute to cognitive decline, especially in SA patients.

In the multivariate regression analysis, BMR appeared to be highly correlated with MoCA score and SMI in two models. This was similar to the research that declared older male patients with SA and frailty have a higher BMR reduction [25]. However, there is less evidence of a relationship between BMR and cognition. Our results gave evidence that BMR might be the key potential factor affecting the link between and progress in MCI and SA. Moreover, the positive relationship of dietary phospholipids, fatty acid and choline on MoCA score and the negative correlation of fat and total fatty acids on SMI provided more evidence that PC-, SM-, fat-, MUFA-, PUFA-, SFA- and PC-related choline played key roles in the health of cognition and muscle.

LASSO and multivariate logistic regression were used to identify the potential predictors and conduct prediction models. As expected, a higher level of BMR was associated with a lower risk of MCI, SA, or MCI and SA. Meanwhile, phosphatidylcholine could reduce the incidence of MCI; C20:1 was associated with a lower risk of SA; and protein and C22:1 showed a lower risk of MCI, SA, and MCI and SA separately. After conducting the ROC model, BMR could significantly enhance the discriminating capability of each prediction model. Further training and validation tests demonstrated that the predicted probability was highly consistent with the actual probability. 

The results shown above provide us a suggestion. The elderly with lower BMR and inadequate intake of PC-related choline and MUFA had a higher risk of MCI and/or SA. BMR is more likely to link to MCI to develop SA, while lack of PC and MUFA make it more likely for SA patients to have cognitive decline. SM, SFA and PUFA might be effective factors as well. In the elderly, pro-inflammatory cytokines, known to be associated with sarcopenia and frailty, could be decreased by consumption of walnuts, characterized by a healthy lipid profile [26].

To further prove the above results, nontargeted and targeted lipidomic analysis were conducted in the erythrocyte of subjects. As expected, the clear discriminations of lipid profile in MCI–control, SA–control, and MCI and SA–control were observed both in nontargeted and targeted lipidomics analysis.

Untargeted results showed PC, PE, SM and TAG were the top changing subclasses in three comparisons. PC and TAG were altered more obviously in all three comparisons. As the key role of dietary PC and negative effects of dietary fat, PC and TAG might be the important altered subclasses in the link between MCI and SA. The changes in PE and SM were found in SA–control and MCI and SA–control, which implied they might be easier to trigger SA.

MRM-targeted results gave us confidence that the downregulation of PEs and SM (26:0) could be considered as prospective biomarkers for higher risk of suffering from MCI and/or SA. These results were highly similar to a nested case–control study in which lower levels of PE and SM are associated with greater odds of cognitive decline in the elderly people of France [27]. Another piece of research into targeted metabolomics provides more evidence that SM in saliva in MCI/AD patients decreases significantly, although there is no statistical significance [28]. Reduced levels of Acyl–Alkyl-PCs in saliva are also indicated to be predictors of MCI and AD [28]. Therefore, PCs were checked in this study next.

A PC with the acyl chain containing ≤ 2 unsaturated double bonds was upregulated in case groups in untargeted tests. This was opposite to the outcome of dietary nutrition. However, PC with more unsaturated double bonds was downregulated in targeted tests. This is similar to other research that shows PC (16:0/20:5), PC (16:0/22:6) and PC (18:0/22:6) (PC with 5–6 bonds) decrease in MCI and AD patients, compared with control [29,30,31], while PC (40:4) and PC (36:3) (PC with 3–4 bonds) increase in AD [32,33]. Furthermore, evidence convinced us that PCs containing more double bonds (such as 16:0/22:6, 18:0/22:6, 18:2/22:6 and 18:1/22:6) decreased in FABP3-overexpressing muscles (FABP3 is recognized as a valuable target for SA), whereas PCs with fewer double bonds (18:1/18:2, 18:0/18:2 and 16:0/16:0) increased [34]. The above results undoubtedly suggested that not only the disordered expressions of PCs but also the attenuation of PCs with more double bonds of acyl chains might be the key link between MCI and SA. It meant these alterations of PCs might be the indicators for higher risk of suffering comorbidities of MCI and SA. Furthermore, the results convinced us that the sources of PCs from different food were important for the elderly. DHA-PC, as the abundant nutrient in marine foods, has been demonstrated as having stronger effects of alleviating age-related memory loss and cognitive deficiency in SAMP8 mice than commercial fish oil and DHA-free PC [35].

The above-mentioned results brought our attention to the type of acyl chain on the different subclasses of lipids. Our results made us consider the EPA as an important factor in the link between MCI and SA. This opinion is also confirmed by the research that showed egg yolk PC (conventional type PC), squid PC (rich in EPA) and sea cucumber PC (rich in DHA) can all diminish the cognitive decline and biological damage, while EPA and DHA partly enhanced the beneficial effects. Moreover, AA, known as an integral constituent of biological cell membrane and a provider of membrane fluidity and flexibility, is necessary for the function of all cells, especially in the nervous system and skeletal muscles [36]. Dietary LA is a precursor for AA that is produced by step-wise desaturation and chain elongation. In the present study, AA-PC and AA-CE were lower in SA and MCI and SA than the control, whereas LA-lipids and LA-CE were upregulated. These opposite results of AA and LA might be associated with the disorder of desaturase and chain elongation in individuals of MCI or SA.

This speculation was also supported by the carbon chain length detection. Not only the upregulation trends of lipids with C16 and C18 but also the downregulation of lipids with C22 and >C24 reminded us that MCI and SA, especially SA, might be involved in a lipid profile remodeling of very-long-chain fatty acids (VLCFA) and long-chain fatty acids (LCFA). It is reported in the research of myoblasts that functional crosstalk between the elongate complex and desaturase is related to the acyl chain length [37]. However, further evidence is required for this new speculation.

From the perspective of dietary nutrition, the results provided more available data and theoretical insight to help prevent cognitive decline and sarcopenia. This study provides novel scientific evidence for prospective biomarkers for a higher risk of cognitive decline and/or SA in elderly populations.

## 5. Conclusions

In conclusion, the inadequate intake and lower concentrations of erythrocyte lipid of PC, SM, PE and unsaturated fatty acids with a lower level of BMR might be the key point of MCI and/or SA. Decreased PC with more unsaturated double bonds, lower lipids with EPA and AA, higher LA and a remodeled length of acyl chain might have a close relationship with the link. They are all prospective biomarkers for higher risk of cognitive decline and/or SA in elderly populations.

## Figures and Tables

**Figure 1 nutrients-14-05321-f001:**
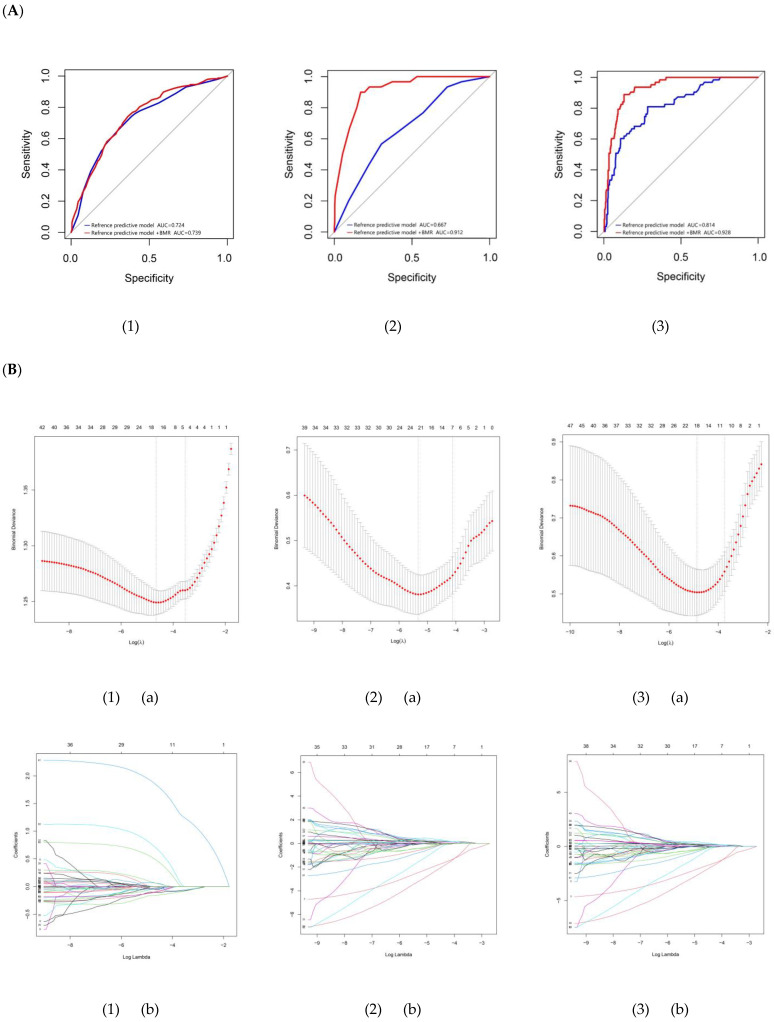
(**A**) Receiver operating characteristic curve analysis for predictive models. (**B**) Variables selection by least absolute shrinkage and selection operator (LASSO) logistic regression. The vertical dotted line points to the optimal lambda value and the number of optimal predictors. (**C**) Receiver operating characteristic curve validation of risk prediction model in the training and validation set. (**D**) Calibration curves of MCI, SA, and MCI and SA in the training and validation set. (**E**) Multivariate logistic regression analyses and nomogram for predicting the MCI, SA, and MCI and SA probability. (1) MCI–Ctrl group; (2) SA–Ctrl group; (3) MCI and SA–Ctrl group. (a) Selection of the parameters in the LASSO model by 10-fold cross-validation based on minimum criteria; (b) the pathway of coefficients among all variables; (c) the training set; (d) the validation set. AUC, area under the curve; BMR, basal metabolic rate; PCho, phosphorylcholine; OR, odds ratio. Adjusted for age, sex, education, BMI, TC, TAG, LDL-C, HDL-C, hypertension, diabetes, dyslipidemia, arthritis. * *p* < 0.05, ** *p* < 0.01. Q1,1st quartile; Q2, 2nd quartile; Q3, 3rd quartile; Q4, 4th quartile.

**Figure 2 nutrients-14-05321-f002:**
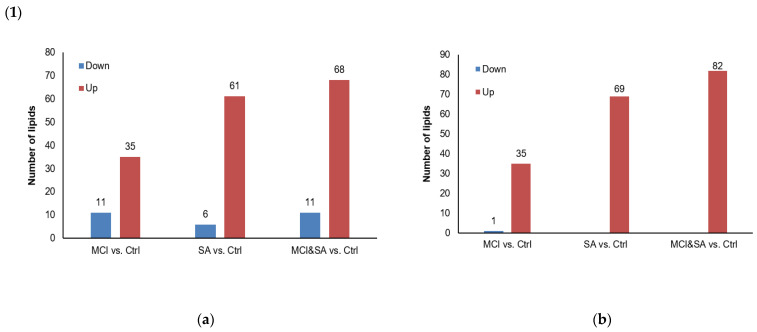
(**1**) Number of lipids with significant changes in different comparisons (*VIP* > 1, *p* < 0.05). (**2**) Number of differential lipid species according to (**1**) in (a) positive and (b) negative ionization mode. CE, cholesteryl ester; Cer, ceramides; DGTS, diacylglyceryltrimethylhomoserine; FA, fatty acids; GlcADG, glucuronosyldiacylglycerol; HexCer, hexosylceramide; LPC, lyso-phosphatidylcholines; PC, phosphatidylcholine; PE, phosphatidylethanolamines; PI, phosphatidylinositol; PS, phosphatidylserine; SM, sphingomyelins; TAG, triglycerides.

**Figure 3 nutrients-14-05321-f003:**
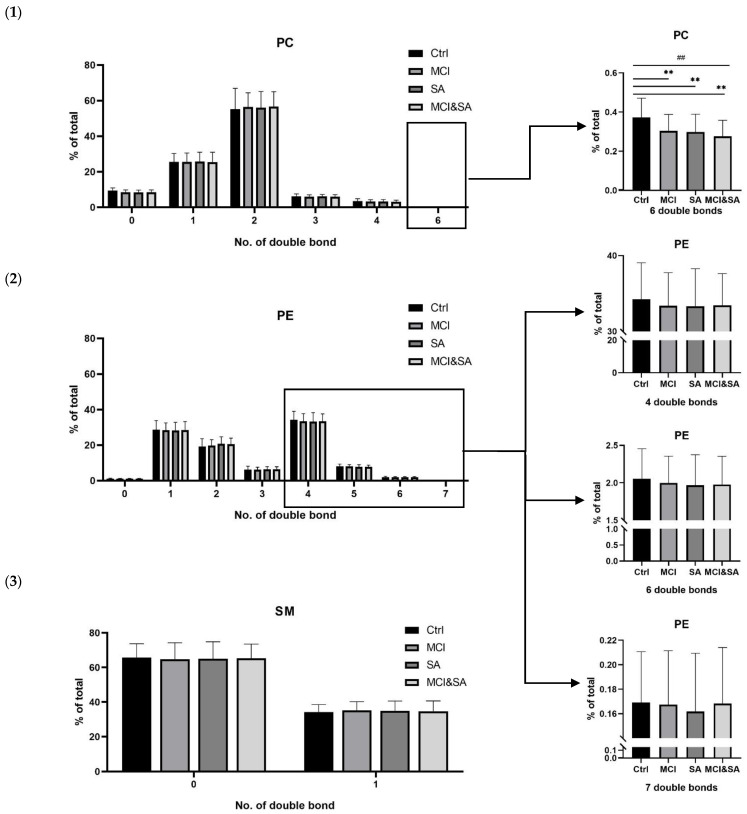
The comparisons for the percentage of (**1**) PC, (**2**) PE and (**3**) SM species contained a different number of double bonds in total lipids based on targeted lipidomic analysis. PC, phosphatidylcholine; PE, phosphatidylethanolamine; SM, sphingomyelin. ** *p* < 0.01, ^##^
*p* < 0.01.

**Table 1 nutrients-14-05321-t001:** Demographic and clinical characteristics of subjects.

	Total	*Categories*				*p* Value
		Ctrl	MCI	SA	MCI & SA	
**Demographic characteristics**						
N	1050	440	490	41	79	
Age	70 (67, 73)	70 (67, 72) ^a,b^	69 (67, 73) ^c,d^	72 (68, 78) ^a,c^	73 (69, 77) ^b,d^	<0.001 **
female, n (%)	629 (59.9%)	291 (66.1%) ^a^	275 (56.1%) ^a^	22 (53.7%)	41 (51.9%)	0.005 **
Education						
Illiterate n (%)	234 (22.3%)	144 (32.7%) ^a^	52 (10.6%) ^a,b,c^	21 (51.2%) ^b,d^	17 (21.5%) ^c,d^	<0.001 **
Primary school n (%)	353 (33.6%)	181 (41.1%) ^a,b^	140 (28.6%) ^a^	13 (31.7%)	19 (24.1%) ^b^	<0.001 **
Junior high school n (%)	376 (35.8%)	81 (18.4%) ^a,b^	256 (52.2%) ^a,c^	5 (12.2%) ^c,d^	34 (43.0%) ^b,d^	<0.001 **
High school and above n (%)	87 (8.3%)	34 (7.7%)	42 (8.6%)	2 (4.9%)	9 (11.4%)	<0.001 **
BMI (kg/m^2^)	25.9 (23.7, 28.2)	26.7 (24.3, 28.9) ^a^	26.3 (24.1, 28.2) ^b^	21.6 (19.8, 23.6) ^a,b^	22.1 (20.4, 23.8) ^a,b^	<0.001 **
Emaciation n (%)	14 (1.3%)	2 (0.5%) ^a,b^	4 (0.8%) ^c^	2 (4,9%) ^a^	6 (7.6%) ^b,c^	<0.001 **
Normal n (%)	278 (26.5%)	90 (20.5%) ^a,b^	103 (21.0%) ^c,d^	30 (73.2%) ^a,c^	55 (69.6%) ^b,d^	<0.001 **
Overweight n (%)	458 (43.6%)	196 (44.5%) ^a,b^	237 (48.4%) ^c,d^	8 (19.5%) ^a,c^	17 (21.5%) ^b,d^	<0.001 **
Obesity n (%)	300 (28.6%)	152 (34.5%) ^a,b^	146 (29.8%) ^c,d^	1 (2.4%) ^a,c^	1 (1.3%) ^b,d^	<0.001 **
BMR (kcal)	1258 (1153, 1387)	1276 (1169, 1396) ^a,b^	1273 (1168, 1426) ^c,d^	1120 (1067, 1249) ^a,c^	1138 (1045, 1282) ^b,d^	<0.001 **
MoCA	21 (17, 23)	22 (20, 25) ^a,b,c^	19 (16, 22) ^a^	20 (15, 24) ^b,d^	18 (12, 21) ^c,d^	<0.001 **
**Chronic diseases**						
Arthritis n (%)	110 (11.6%)	42 (10.4%) ^a^	49 (11.1%) ^b^	4 (11.4%)	15 (22.7%) ^a,b^	0.034 *
Hypertension n (%)	621 (65.6%)	279 (69.1%)	285 (64.5%)	22 (62.9%)	35 (53.0%)	0.068
Diabetes n (%)	215 (22.7%)	91 (22.5%)	105 (23.8%)	7 (20.0%)	12 (18.2%)	0.753
Dyslipidemia n (%)	352 (34.5%)	151 (35.2%)	165 (34.7%)	9 (22.0%)	27 (36.5%)	0.379
**Serum Cholesterol**						
N	1020	429	476	41	74	
TC (mmol/L)	4.659 ± 0.975	4.690 (3.970, 5.290)	4.565 (3.943, 5.228)	4.610 (4.035, 5.100)	4.825 (4.068, 5.493)	0.284
TAG (mmol/L)	1.310 (0.940, 1.868)	1.340 (0.990, 1.915)	1.320 (0.943, 1.865)	1.040 (0.770, 1.490)	1.160 (0.875, 1.595)	0.014 *
HDL-C (mmol/L)	1.250 (1.070, 1.450)	1.250 (1.090, 1.450)	1.240 (1.050, 1.430)	1.300 (1.095, 1.590)	1.230 (1.018, 1.603)	0.207
LDL-C (mmol/L)	3.007 ± 0.893	3.005 ± 0.910	2.972 ± 0.881	3.076 ± 0.730	3.209 ± 0.939	0.189
**Dietary intakes**						
N	975	407	457	35	76	
Energy (kcal/d)	1781 (1424, 2155)	1841 (1467, 2208)	1750 (1364, 2128)	1925 (1574, 2345)	1741 (1365, 2044)	0.031 *
Protein (g/d)	59.0 (44.0, 75.2)	61.6 (47.5, 79.2) ^a,b^	58.1 (41.9, 72.1) ^a^	65.5 (40.2, 86.8) ^c^	52.5 (37.5, 65.3) ^b,c^	<0.001 **
CHO (g/d)	202.0 (155.0, 264.8)	211.3 (159.4, 285.1) ^a^	197.7 (152.4, 253.5) ^a^	204.1 (156.6, 271.8)	185.8 (148.4, 241.8)	0.015 *
Fat (g/d)	75.8 (58.1, 95.6)	75.7 (59.9, 96.8)	75.3 (57.4, 94.1)	92.9 (72.8, 107.2)	79.1 (53.0, 100.4)	0.069
Cholesterol (mg/d)	333.3 (204.5, 397.5)	347.5 (225.7, 405.0)	329.8 (192.3, 390.7)	361.7 (294.2, 450.4) ^a^	298.4 (146.9, 373.9) ^a^	0.004 **
PC (mg/d)	82.610 (59.586, 101.892)	85.472 (62.950, 106.057) ^a^	80.987 (58.946, 100.311)	86.742 (75.755, 107.382) ^b^	69.074 (45.724, 95.031) ^a,b^	0.001 **
SM (mg/d)	4.136 (2.917, 5.536)	4.356 (3.020, 5.876) ^a^	3.969 (2.889, 5.403)	4.780 (3.437, 5.733)	3.417 (2.313, 5.248) ^a^	0.003 **
Total fatty acid (g/d)	68.677 (53.198, 87.310)	68.253 (53.575, 87.361)	68.222 (52.203, 86.382)	85.462 (66.905, 99.918)	72.251 (47.977, 91.911)	0.061
SFA (g/d)	19.029 (13.081, 24.502)	19.013 (13.698, 25.468)	18.730 (12.988, 23.976)	21.590 (14.421, 29.010)	18.308 (12.175, 23.532)	0.027 *
MUFA (g/d)	13.168 (9.404, 17.986)	13.641 (10.290, 18.947) ^a^	13.014 (8.776, 17.355)	15.523 (10.026, 20.032)	11.160 (8.146, 15.677) ^a^	0.001 **
PUFA (g/d)	4.908 (3.164, 7.836)	5.154 (3.376, 8.022) ^a^	4.874 (3.078, 7.797) ^b^	6.268 (3.732, 8.734) ^c^	3.776 (2.392, 6.318) ^a,b,c^	0.001 **
Betaine (mg/d)	110.0 (78.2, 157.8)	121.7 (84.0, 168.1) ^a,b^	107.3 (78.9, 145.8) ^a^	108.5 (66.5, 139.2)	86.5 (64.4, 142.1) ^b^	<0.001 **
GPC (mg/d)	7.769 (5.656, 10.420)	8.142 (6.057, 11.190) ^a^	7.737 (5.596, 10.164) ^b^	6.972 (5.381, 10.278)	6.251 (4.622, 9.184) ^a,b^	<0.001 **
Phosphatidylcholine (mg/d)	2.450 (1.685, 3.297)	2.627 (1.848, 3.391) ^a,b^	2.381 (1.681, 3.237) ^a,c^	2.674 (1.714, 3.615)	1.848 (1.337, 2.901) ^b,c^	<0.001 **

BMI, body mass index; BMR, basal metabolic rate; CHO, carbohydrate; GPC, glycerol phosphatidylcholine; HDL-C, high-density lipoprotein cholesterol; LDL-C, low-density lipoprotein cholesterol; MoCA, Montreal cognitive assessment score; MUFA, monounsaturated fatty acid; PC, lecithin; PUFA, polyunsaturated fatty acid; SFA, saturated fatty acids; SM, sphingomyelin; TAG, triglyceride; TC, total cholesterol. Not all sample size of indexes were 1050 because of deficiency. Each number of percentages was calculated by their own sample size. ^a,b,c,d^ Means with the same upper letter (a/b/c/d) in the same line are significantly different at *p* < 0.05. * *p* < 0.05. ** *p* < 0.01 in all groups.

**Table 2 nutrients-14-05321-t002:** Multiple linear regression models predicting MoCA score and SMI.

	Model1				Model2			
	MoCA		SMI		MoCA		SMI	
	B	*p* Value	B	*p* Value	B	*p* Value	B	*p* Value
BMR (kcal)	0.004	0.002 **	0.005	0.007 **	0.002	<0.001 **	0.002	<0.001 **

BMR (kcal)	0.004	0.018 *	0.002	<0.001 **	0.004	0.015 *	0.002	<0.001 **
Energy (kcal/d)	0.001	0.002 **	−0.000	0.095	0.001	0.039 *	−0.000	0.040 *

BMR (kcal)	0.003	0.030 *	0.002	<0.001 **	0.003	0.043 *	0.002	<0.001 **
Protein (g/d)	0.037	0.001 **	0.002	0.008 **	0.044	0.001 **	0.002	0.138
CHO (g/d)	−0.001	0.672	−0.000	0.040 *	−0.002	0.593	−0.000	0.314
Fat (g/d)	−0.013	0.032 *	−0.002	0.001 **	−0.022	0.004 **	−0.002	0.001 **

BMR (kcal)	0.004	0.012 *	0.002	<0.001 **	0.004	0.017 *	0.002	<0.001 **
Cholesterol (mg/d)	0.002	0.014 *	−0.000	0.544	0.003	0.012 *	−0.000	0.332

BMR (kcal/d)	0.004	0.018 *	0.002	<0.001 **	0.004	0.021 *	0.002	<0.001 **
PC (mg/d)	0.012	0.002 **	−0.000	0.753	0.014	0.002 *	−0.000	0.292

BMR (kcal)	0.004	0.014 *	0.002	<0.001 **	0.004	0.016 *	0.002	<0.001 **
SM (mg/d)	0.140	0.019 *	0.000	0.978	0.156	0.023 *	−0.007	0.238

BMR (kcal)	0.004	0.009 **	0.002	<0.001 **	0.004	0.010 *	0.002	<0.001 **
Total fatty acid (g/d)	0.003	0.600	−0.001	0.019 *	−0.002	0.746	−0.002	0.003 *

BMR (kcal)	0.004	0.013 *	0.002	<0.001 **	0.004	0.015 *	0.002	<0.001 **
SFA (g/d)	0.038	0.034 *	−0.001	0.489	0.032	0.126	−0.003	0.094

BMR (kcal)	0.003	0.023 *	0.002	<0.001 **	0.004	0.025 *	0.002	<0.001 **
MUFA (g/d)	0.077	<0.001 **	0.001	0.503	0.077	0.002 **	−0.001	0.668

BMR (kcal)	0.003	0.020 *	0.002	<0.001 **	0.004	0.023 *	0.002	<0.001 **
PUFA (g/d)	0.106	0.001 **	0.002	0.58	0.095	0.012 *	0.000	0.970

BMR (kcal)	0.003	0.019 *	0.002	<0.001 **	0.004	0.018 *	0.002	<0.001 **
Betaine (mg/d)	0.008	<0.001 **	−0.000	0.983	0.008	0.001 **	−0.000	0.604

BMR (kcal)	0.004	0.020 *	0.002	<0.001 **	0.004	0.019 *	0.002	<0.001 **
GPC (mg/d)	0.091	0.007 **	0.003	0.333	0.100	0.010 *	−0.002	0.621

BMR (kcal)	0.003	0.046 *	0.002	<0.001 **	0.004	0.030 *	0.002	<0.001 **
Phosphatidylcholine (mg/d)	0.549	<0.001 **	−0.000	0.987	0.466	0.001 **	−0.006	0.599

BMR (kcal)	0.001	0.423	--	--	0.002	0.299	--	--
SMI (kg)	1.282	0.001 **	--	--	1.153	0.015*	--	--

BMR (kcal)	--	--	0.002	<0.001 **	--	--	0.002	<0.001 **
MoCA	--	--	0.009	0.001 **	--	--	0.008	0.018 *

BMR, basal metabolic rate; CHO, carbohydrate; GPC, glycerol phosphatidylcholine; MoCA, Montreal cognitive assessment score; MUFA, monounsaturated fatty acid; PC, lecithin; PUFA, polyunsaturated fatty acid; SFA, saturated fatty acids; SM, sphingomyelin; SMI, skeletal muscle index. Model1 adjusted for age, sex, education, BMI, arthritis and TAG. Model2 adjusted for the variables in model 1 and PASE score. * *p* < 0.05. ** *p* < 0.01.

**Table 3 nutrients-14-05321-t003:** Multivariate logistic regression analyses for three models.

	Q1	Q2		Q3		Q4		*p* Value
		OR (95%CI)	*p* Value	OR (95%CI)	*p* Value	OR (95%CI)	*p* Value	
MCI vs. Ctrl ^a^								
BMR (kcal)	Ref	1.175 (0.745, 1.852)	0.488	0.580 (0.364, 0.923)	0.022 *	0.671 (0.421, 1.07)	0.094	0.008 **
Phosphatidylcholine (mg/d)	Ref	0.978 (0.624, 1.534)	0.922	0.552 (0.354, 0.860)	0.009 **	0.569 (0.364, 0.890)	0.013*	0.006 **
SA vs. Ctrl ^a^								
BMR (kcal)	Ref	0.021 (0.002 0.242)	<0.001 **	0.002 (0.000, 0.027)	<0.001 **	<0.001 (0.000, 0.005)	<0.001 **	<0.001 **
C20:1 (g/d)	Ref	0.051 (0.004, 0.608)	0.019 *	1.075 (0.321, 3.606)	0.907	2.293 (0.767, 6.854)	0.137	0.017 *
MCI and SA vs. Ctrl ^a^								
BMR (kcal)	Ref	0.038 (0.008,0.184)	<0.001 **	0.009 (0.001,0.059)	<0.001 **	<0.001 (0,0.003)	<0.001 **	<0.001 **
Protein (g/d)	Ref	1.538 (0.608,3.891)	0.363	0.794 (0.306,2.065)	0.637	0.185 (0.053,0.653)	0.009 **	0.011 *
C22:1 (g/d)	Ref	2.905 (0.916,9.215)	0.070	2.543 (0.716,9.028)	0.149	8.249 (2.545,26.733)	<0.001 **	0.003 **

BMR, basal metabolic rate; OR, odds ratio. ^a^ Adjusted for age, sex, education, BMI, TC, TAG, LDL-C, HDL-C, hypertension, diabetes, dyslipidemia, arthritis. * *p* < 0.05. ** *p* < 0.01.

## Data Availability

Data are available upon reasonable request.

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
