# Peer review of "Correlation between Mild Cognitive Impairment and Sarcopenia: The Prospective Role of Lipids and Basal Metabolic Rate in the Link"

_nutrients, 2022, doi:10.3390/nu14245321_

Round 1

Reviewer 1 Report

Knowing the relationship between mild cognitive impairment and sarcopenia, the paper provides a comprehensive analysis of the lower dietary intake of different phospholipids, unsaturated fatty acids, and choline, correlated to lower levels of basal metabolic rate, concluding that these factors could be biomarkers for the development of mild cognitive impairment and sarcopenia and their link.

The subject is important, the manuscript is well written, and, given the complexity involved, the authors made important contributions. There are some details that need to be attended to:

Lines 63-65: “a phenomenon was found that MCI individuals were more likely to be troubled by decreased muscle mass or strength.” should be deleted

Line 70: Methods - if the methods are not original, they need reference (here and in Supplementary Materials)

Lines 72-78: please revise English in the paragraph

Line 80: AWGS should be defined

Lines 111-119: Statistical analysis from Supplementary Materials can be transferred here 

Line 140: check English in the paragraph

Lines 150-154: this is not very clear

Lines 299-300: sentence needs reference

Lines 335-338: the following idea can be added - “in the elderly, pro-inflammatory cytokines, known to be associated to sarcopenia and frailty, could be decreased by consumption of walnuts, characterized by a healthy lipid profile (linoleic and alpha-linolenic acids) (doi: 10.3390/antiox11071412).

Please check English throughout.

Author Response

Response to Reviewer 1 Comments

Q1: Line 70: Methods - if the methods are not original, they need reference (here and in Supplementary Materials)

Answer: This is our mistake and thank you for your seriousness. Necessary reference have been added in the text.

Q2: Lines 72-78: please revise English in the paragraph

Answer: Thank you very much for your conscientiousness. It has been revised in the paragraph.

Q3: Line 80: Thank you very much for your kindness. AWGS should be defined

Answer: Thank you very much for your preciseness and seriousness. AWGS has been defined in the text.

Q4: Lines 111-119: Statistical analysis from Supplementary Materials can be transferred here

Answer: Thank you very much for your conscientiousness. Statistical analysis has been transferred from Supplementary Materials.

Q5: Line 140: check English in the paragraph

Answer: Thank you very much for your patience. English in the paragraph has been checked.

Q6: Lines 150-154: this is not very clear

Answer: Thank you very much for your kindness. The results have been revised in the text.

Q7: Lines 299-300: sentence needs reference

Answer: Thank you very much for your preciseness. The references have been added there.

Q8: Lines 335-338: the following idea can be added - “in the elderly, pro-inflammatory cytokines, known to be associated to sarcopenia and frailty, could be decreased by consumption of walnuts, characterized by a healthy lipid profile (linoleic and alpha-linolenic acids) (doi: 10.3390/antiox11071412).

Answer: Thank you very much for your good advice. We have carefully read the literature you recommended. The research is a very excellent work which provide the great perspectives. The valuable points inspire us very much, which have been added in the discussion. Thanks a lot.

Q9: Please check English throughout.

Answer: Thank you very much for your good advice. Language editing and proofreading have been done by professor of English language from Capital Medical University again.

Reviewer 2 Report

This fascinating paper widely explores the correlation between dietary intake and lipid metabolism in a sample of 1050 elderly divided into four groups: control, MCI, SA, and MCI + SA.

The results showed that preserved cognition correlates with higher skeletal muscle index and basal metabolic rate; in all three comparisons, they found upregulation of TAG, PC, PS, PE, SM, and Cer. The authors also investigated the dietary intake correlation within the groups and found decreased phospholipids and fatty acid intake.

The methodology is very comprehensive and well-written.

The discussion could be refined by comparing these results to existing literature (only a few papers are cited in the debate). Finally, the authors could discuss the clinical implication of their work.

A few language errors should be addressed.

Figures could be made more clear to increase readability (es. nomogram).

Author Response

Response to Reviewer 2 Comments

Q1: The discussion could be refined by comparing these results to existing literature (only a few papers are cited in the debate). Finally, the authors could discuss the clinical implication of their work.

Answer: Thank you very much for your conscientiousness. Discussion has been improved in the text.

Q2: A few language errors should be addressed.

Answer: Thank you very much for your good advice. Language editing and proofreading have been done by professor of English language from Capital Medical University again.

Q3: Figures could be made more clearer to increase readability (es. nomogram).

Answer: Thank you very much for your preciseness. We have provided clearer original pictures. If the clarity is not enough, please contact us directly, and we will try our best to solve it. Thanks a lot.
